# Microfluidic Liver-on-a-Chip for Preclinical Drug Discovery

**DOI:** 10.3390/pharmaceutics15041300

**Published:** 2023-04-21

**Authors:** Jingyu Fu, Hailong Qiu, Cherie S. Tan

**Affiliations:** 1Academy of Medical Engineering and Translational Medicine, Tianjin University, Tianjin 300072, China; 2Tianjin Key Laboratory of Functional Crystal Materials, Institute of Functional Crystal, Tianjin University of Technology, Tianjin 300384, China

**Keywords:** drug discovery, liver-on-a-chip, organ-on-chip, microfluidics

## Abstract

Drug discovery is an expensive, long, and complex process, usually with a high degree of uncertainty. In order to improve the efficiency of drug development, effective methods are demanded to screen lead molecules and eliminate toxic compounds in the preclinical pipeline. Drug metabolism is crucial in determining the efficacy and potential side effects, mainly in the liver. Recently, the liver-on-a-chip (LoC) platform based on microfluidic technology has attracted widespread attention. LoC systems can be applied to predict drug metabolism and hepatotoxicity or to investigate PK/PD (pharmacokinetics/pharmacodynamics) performance when combined with other artificial organ-on-chips. This review discusses the liver physiological microenvironment simulated by LoC, especially the cell compositions and roles. We summarize the current methods of constructing LoC and the pharmacological and toxicological application of LoC in preclinical research. In conclusion, we also discussed the limitations of LoC in drug discovery and proposed a direction for improvement, which may provide an agenda for further research.

## 1. Introduction

Drug development is a long, costly, and highly uncertain process that usually takes 10–15 years [1]. After the preclinical trial, 90% of the approved drugs may still fail during human clinical trials or FDA approval [2,3]. It is also worth noting that these data do not include therapeutic candidates in the preclinical stage; otherwise, the drug discovery failure rate will be even higher. Statistics show that the average development cost of a new drug is around two billion dollars [4]. Most drug failures in clinical trials because the existing models in preclinical development are not appropriate for predicting new compound toxicity or metabolism [5]. The commonly used preclinical models include in vitro cell culture and animal models. Animal models, such as the mice models, are not necessarily effective in predicting human efficacy and toxicity due to nonnegligible interspecies differences in drug targets and metabolism pathways [6,7,8,9,10]. Furthermore, the cost and ethical concerns also require attention. Therefore, in vitro cell culture systems have been widely used as substitutes for animal models [10,11,12]. However, in vitro cell models also have limitations, such as the lack of an extracellular matrix and the inability to pursue pathophysiological studies. For example, primary human hepatocytes (PHHs), the gold standard of toxicity testing, will rapidly depolarize in vitro, and the drug clearance rate will also be significantly reduced unless they are cultured in a particular environment, such as collagen sandwich culture [13]. Engineers are using engineering tools to control the cellular microenvironment, such as semiconductor industrial tools, miniaturization tools, and integration with organs on chips, to improve the stability of PHHs’ functions [14].

One of the primary areas of unfavorable medication effects is the liver, which also performs a significant role in drug metabolism. Thus, the liver can play a major part in assessing drug efficiency and safety in the preclinical stage. Consequently, establishing an in vitro liver model reproducing the liver microenvironment is conducive to understanding the preclinical research of drug development. The recently developed liver-on-a-chip (LoC) technology is innovative in managing the liver microenvironment in vitro, which allows the dynamic flow of cell medium over the cell environment with good robustness and reproducibility. The LoC system can also be integrated into the laboratory as preclinical tests to predict in vitro human experiments [15]. Multiple liver chip platforms have been designed to model drug metabolism [16,17,18,19], hepatotoxicity [7,20,21,22], drug–drug interactions [16,23], cancer [24], and inflammation [18]. The LoC system can also explore how the internal structure of the liver, such as the hepatic vascular system, is affected by oxygen and nutrient gradients, thereby simulating chronic liver disease in vitro [25,26] and searching for candidate drugs for the disease. This review introduces the liver’s physiological microenvironment, focusing on its cell composition and special functions. Then it summarizes the current LoC construction methods using microfluidic technology, including static platform, 3D printing, and perfusion. We then introduce the application of LoC in preclinical development involving pharmacology, toxicology, and drug–drug interactions. Finally, we point out the limitations of the LoC platform and then attempt to suggest future improvements.

## 2. Liver-on-a-Chip

### 2.1. Physiological Microenvironment of the Liver

As the largest internal organ, the liver plays a crucial role in regulating the homeostasis of the internal environment, the anabolism of various substances, and the detoxification of endogenous and exogenous substances, especially in drug metabolism [27,28]. The hepatic artery and hepatic vein provide a double blood supply for rich hepatic sinusoids and delicate hepatic lobule structure. Liver parenchymal cells (hepatocytes), non-parenchymal cells (Kupffer cells, hepatic stellate cells, macrophages, and sinusoid endothelial cells), the extracellular matrix, and the nervous system constitute the complex liver microenvironment. Distinct types of cells form complicated transmission networks and metabolic environments through autocrine and paracrine signal transduction [29,30].

Drug metabolism can be divided into oxidation, reduction, hydrolysis, and conjugation. Phase I metabolism includes oxidation, reduction, and hydrolysis, while phase II metabolism includes conjugation. Hepatocytes are rich in various enzymes required for phase I and phase II drug metabolism. Among them, cytochrome P450 (CYP450) enzymes are terminal oxygenases involved in nearly all phase I drug metabolism steps in the liver. CYP450 enzymes have apparent species differences, leading to many differences in metabolic pathways and metabolites of drugs between animals and humans. Polymorphisms are a critical feature of CYP450 and the fundamental reason for individual differences in drug response. In addition, CYP450 can be induced and inhibited; that is to say, drugs or other exogenous chemicals will alter the number and activity of CYP450, which may result in metabolic drug interactions.

Hepatocytes, known as parenchymal cells, account for about 60% of liver cells [31]. Hepatocytes are responsible for the liver’s metabolic activity with rich mitochondria, endoplasmic reticulum, ribosomes, and many enzymes, including CYP450 [32,33]. The normal hepatocytes have polarization: the cell membrane of hepatocytes is divided into the basolateral domains facing the liver sinus and the apical domains facing the bile duct [13].

On the micro-scale, the liver is a reticular structure with hepatic lobules as the functional liver unit. Lobules take the hepatic vein as the center and distribute layers of liver plates around. The liver sinusoid is the lacuna between adjacent liver plates with strong permeability. Blood is supplied to each lobule from hepatic arterioles and portal veins (Figure 1C). Hepatocytes, which are deeply embedded in the endothelium, constitute the parenchyma of the liver. The medication molecules in blood can rapidly diffuse with the aid of sinusoidal endothelial cells, completing the material exchange with hepatocytes [34,35]. Hepatic stellate cells are located in the space of Disse between endothelium and hepatocytes. The Kupffer cells, which facilitate antigen detection and intercellular communication, line the inside of the sinusoid. At last, the sinusoid’s interior is lined with Kupffer cells, facilitating intercellular communication and antigen detection (Figure 1A). Different levels of hepatotoxicity are affected by liver zoning. Low oxygen content will increase CYP activity, leading to increased hepatotoxicity, while locations with low CYP activity will cause less cell harm (Figure 1D).

### 2.2. Liver-on-a-Chip Technology

As mentioned above, drug development requires in vitro liver models to investigate drug metabolism and other aspects. Typical models include 2D planar primary hepatocytes [36,37], 3D printed liver tissue [38,39,40,41], hepatocyte organoids [42], and liver-on-a-chip [43,44,45,46].

Traditional 2D culture of hepatocytes is easy to handle with low cost and plays an essential role in drug research, development, and cell biology. However, liver functions cannot be wholly reproduced because most hepatocytes will soon lose internal biochemical clues and cell–cell communication, leading to phenotypic changes. The rapid development of 3D printing liver tissue and constructing liver organoid technologies provides a viable strategy for in vitro liver models; however, they are also limited by insufficient accuracy or lack of spatial structure.

With the ability to precisely control the culture environment at the micro-scale, microfluidics-based cell culture technologies have recently captured the most attention. It is assayed robust and replicates the hepatic microenvironment. Moreover, with the recent rise of nanomedicine in the application of vaccines, the LoC system can conduct abundant and accurate research on the toxicity of nanomedicines [47], making up for the lack of nanotoxicity assessment. In addition, the hepatocytes cultivated in these devices under flowing circumstances allow for more frequent nutrition and waste exchange than in traditional methods. In contrast, LoC manufacturing calls for more complex operations.

#### 2.2.1. Cell Sourcing

The reliability of the simulation effect of liver models is significantly affected by the source of the cells. Primary human hepatocytes (PHHs), primary animal cells, stem cell-derived hepatocytes, and immortalized liver cell lines are the main cell types used for liver tissue reconstruction in vitro.

PHHs with good reproducibility in vitro maintained the metabolic function of the liver, mainly retained the enzyme level consistent with that in vivo, becoming the “gold standard” for drug tests and widely used in drug metabolism and toxicology research [48,49,50]. In addition, PHHs with population heterogeneity are essential to reflect in drug testing [51]. However, culturing PHHs on a medium presents a slice of problems, such as the rapid loss of functionality and polarity in vitro [31,52,53,54]. Therefore, researchers have tried changing the culture materials, co-culturing, 3D culturing, and other methods but failed to establish a simple, efficient, and stable culture system of PHHs in vitro. Moreover, PHHs have constraints such as high costs resulting from donor shortage and variability.

Animal-based testing has built an essential bridge between in vitro drug research and clinical trials, which is critical in drug safety research. In addition, since research animals are easily accessible commercially, they can be employed as sources of fresh primary cells without the limitation of donor shortage [55,56]. However, they have significant interspecies differences in drug targets, the magnitude of hepatic first-pass metabolism, and other aspects. Especially the distinct variations in CYP450 induction by medications between humans and animals (rats, dogs) have been well established, obstructing their wide application [57].

Immortalized liver cell lines, which can be manipulated with unlimited proliferation by inhibiting cellular senescence [58,59], are widely used in drug toxicology research in vitro. Standard liver cell lines are derived from hepatocellular carcinoma (HCC) [60,61], such as Huh7, HepG2, and HepaRG. HepG2 cells have been used in toxicology and pharmacology since the 1970s. They have diverse differentiation functions and stable phenotypes, but the CYP450 expression level is not as high as that of HepaRG cells when HepaRG cells differentiate into hepatocytes [62,63]. However, compared with PHHs, the cell lines generally have lower activities of CYP450 and detection accuracy in drug toxicity.

Hepatocytes derived from stem cells have emerged as a new source of liver cells in recent years [64]. Regarding function, stem cells have the potential for self-renewal and multiple types of differentiation. They are the most primitive cells in the cell line, which can differentiate into cells of a particular tissue type in vivo [65]. Common stem cell lines include embryonic stem cells (ESCs) and induced pluripotent stem cells (iPSCs), which can differentiate into various liver cells [66,67] and hematopoietic stem cells [68,69] with stable metabolic function and specific gene expression ability. The iPSCs have resembled sensitivity as PHHs in detecting the degree of drug-induced liver injury [70]. The culture medium is supplemented with growth factors or other substances to carry out this differentiation. Differentiated cells can retain the donor’s genotype to study personal effects, so there is no need to biopsy the target organ itself. Accordingly, the manipulation of stem cells is remarkably more complex than that of liver cell lines. When compared to the organoids developed by PHHs, hepatocytes generated from iPSCs consistently secrete less albumin, have much lower CYP450 activity, and express immature markers.

Overall, the physiological culture conditions of the liver models are strongly influenced by the source and kind of cells employed, such as the characteristics of primary hepatocytes, HepG2, and HepaRG cell lines mentioned above. In addition, many cells require sensitive environments, so rational in vitro liver model systems must provide these specialized environments in culture. Therefore, currently, the researchers turn to microfluidics to effectively control the culture microenvironment.

#### 2.2.2. Microfluidic Strategies for Liver-on-a-Chip

Microfluidics is a downsizing of biological separation and analysis techniques with a high surface-to-volume ratio, automation, and integration [71,72]. The cells in conventional culture methods are in a semi-static environment, and the chemicals added in the laboratory can only affect the cells through diffusion, which is much of a difference compared with the cells exposed to complex chemical and physical stimuli in vivo. Researchers used microfluidic devices to provide oxygen, shear stress, and other conditions for cell culture in vitro, pursuing better simulation of the physiological functions of tissue and organs [73]. This technology became an interdisciplinary field of research in the 1990s, combining micro-device technologies, chemical sensor technology, and analytical chemistry. These chips recreate three essential characteristics of tissues or organs: (i) biochemical microenvironment, which includes chemokine, growth factor, and nutrient gradients; (ii) 3D microarchitecture or the spatial distribution of various cell types; and (iii) mechanical microenvironment, which includes mechanical compression, cyclic strain, and shear stress [5,74,75,76,77] (Figure 1B).

The initial static culture platforms inspired perfused devices to culture primary hepatocytes in vitro. In the first LoC reported in 2007 [78], microfluidic technology was used to simulate the endothelial barrier in the hepatic sinuses to protect liver cells from damage due to high shear stress (Figure 2B). This method allowed both human and rat primary hepatocytes to maintain more than 90% cellular viability after seven days. In 2008, Khetani et al. [11] designed a platform that showed the importance of co-culture and spatial structuring. Experiments used a 24-well polydimethylsiloxane (PDMS) elastic mold to provide a culture environment, where primary hepatocytes were selectively adhered to matrix protein-coated regions and surrounded by mouse 3T3-J2 fibroblasts (Figure 2A). Gravity is the primary source of propulsion for this basic perfusion platform, which is easy to use and eliminates the need for external power supplies. This is a widely adopted platform in the drug metabolism area of drug discovery, whereas the shear stress and nutrient supply cannot be constantly controlled. The pumped flow can achieve this control and improve the consistency of supply. Pump-driven 2D platforms can replicate zonation features thanks to the chips’ intricacies [79]. The gradient generator has a typical Christmas-tree structure, creating a concentration gradient between the two extremities at the entrance and connecting to the cell culture chamber (Figure 2C). Cell culture chambers were made as an elongated hexagon shape to simulate the hepatic lobular architecture. This perfusion device provided an acceptable low-shear-stress environment for hepatocytes to achieve more stable function in vitro. These similar membrane-based 2D platforms are widely used in hepatotoxicity assessment [80] and disease modeling [81]. Compared with the 2D platform, the cells in the 3D culture system can be closely connected without affecting drug diffusion, such as the 3D spherical cell culture. The long-term stability of the human liver spheroids system makes it an effective tool for predicting the hepatic clearance of drugs in vivo. A single spheroid can only contain a few cells, and the direct pooling of spheroids will lead to the fusion of spheroids, causing central necrosis or deposition of the bottom and loss of three-dimensional structure. The microporous array processed by the low-adhesion layer was used to improve the pooling of multiple spheroids [82,83], and successfully generated up to 100 spheroids (Figure 2D). Nevertheless, such spheroids cannot be compared to the organoids produced by differentiated stem cells in terms of physiological correlation. In addition to the technologies described above, 3D bioprinting, layer-by-layer deposition, cell microarrays, and many other strategies can be used to construct liver chips.

These rich designs and formats all have their strengths and weaknesses. In the experimental design for drug development, the focus should be on selecting the critical parameters needed, simplifying other aspects of the need. It is essential to seek clues from the in vivo liver environment and to be good at incorporating different types of platforms to achieve better results.

## 3. Application in Preclinical Studies of Drug Discovery

Preclinical research of drug development refers to the safety and effectiveness testing of new active substances before human trials. The goal is to identify and exclude ineffective, toxic, or unqualified candidate drugs at each stage. In preclinical research, the significance of the LoC platform is to reduce or replace conventional mammalian models and conduct PK/PD prediction and toxicological safety research closer to the natural human environment.

### 3.1. Pharmacokinetics and Pharmacodynamics

Currently, single-organ and multi-organ liver models are employed in drug metabolism research. The single LoC model focuses on the drug ADME (absorption, distribution, metabolism, and excretion) in the only liver. The multi-organ model examines how drugs react when different organs in the body interact with the liver, including how much of a drug is present in various organs and how metabolites affect other organs. The basis for the drug ADME is to improve the survival time and functional activity of cells in the LoC system. The widely accepted markers are albumin synthesis and urea secretion [84]. It has been proven that the activity of hepatocytes in vitro can be significantly improved in co-culture conditions. Almost all liver physiological systems in vitro used for drug research have chosen the co-culture mode, regardless of 2D or 3D, and will change co-cultured cells based on research purposes, such as immune-related cells or inflammatory-related cells [85]. After ensuring the cell activity, the LoC system needs to meet the drug metabolism function, which is the expression induction ability of drug metabolism enzymes (such as CYP) [86]. The prolonged survival time of cells within the system is conducive to the induction and long-term dependent inhibition of drug-related enzymes, thereby more realistically reducing the reaction process of drugs in the human body. Using typical substrates of various drug-metabolizing enzymes as probes to study metabolic processes is very common and easy to operate. Compared to animal models, the LoC system can monitor metabolic processes in real-time and transparently and flexibly adjust the physiological conditions of the system. Finally, the drug clearance process simulation compares the amount and distribution of metabolites in the LoC system with human urine. The LoC system analyzes the clearance efficiency and accumulated toxicity of metabolites of candidate drugs to determine important targets such as drug bioavailability, in vivo half-life, and dosing interval. With complete and stable functions, the LoC system can play a role in predicting the metabolic process of clinical drugs and the acceptable dose of drugs to the human body.

The metabolism of drugs will be affected by inflammation. Severe inflammation will cause liver damage and even endanger life. Sarkar et al. [18] designed a 3D perfusion liver model with co-cultured hepatocytes and Kupffer cells, which can release inflammatory factors, to examine the anti-inflammatory effect of glucocorticoids on the liver. Compared with a traditional 2D single culture, both 3D perfusion and co-culture models of hepatocytes and non-parenchymal cells were conducive to long-term stable liver function in vitro. After the inflammation was induced by lipopolysaccharide in the culture environment, oxycortisone in the model showed first-order kinetic characteristics of metabolism. Its clearance value and metabolites are generally related to human data, demonstrating the effectiveness of this 3D perfusion platform in screening anti-inflammatory compounds. Long et al. [16] carried out a similar study with the same 3D perfusable human liver co-culture platform (Figure 3A,B). This culture mode prolonged the time for hepatocytes to maintain functional activity in vitro so that the inhibition of chronic inflammatory factor chronic interleukin 6 (IL-6) on the activity of metabolic enzyme cytochrome P450 3A4 isoform (CYP3A4) could be detected, which occurred within two weeks. Under the influence of inflammatory factors and therapeutic antibodies, changes in CYP3A4 activity were observed in vitro, which was revealed through the metabolism of small molecule drugs. The long-term stability of this model realized its prediction function of drug–drug interaction, which is of great significance in the preclinical drug development stage.

In addition to predicting drug metabolism at the individual level, variability in the population can also be considered. Tsamandouras et al. [19] applied the LoC platform in quantitative pharmacological research and analyzed hepatocytes from different donors. A three-dimensional perfusion human liver micro-physiological system can prolong the survival of cells, and alleviate the rapid loss of metabolic function, providing suitable conditions for studying low-clearance compounds in drug metabolism. The author used this model to explore the metabolic efficiency of six compounds. After statistical processing of the data obtained from the LoC platform, it was applied to the modeling and simulation framework to transfer the results of population variability in vitro to in vivo. Using modeling and simulation for more accurate data evaluation is effective. Docci, Luca et al. [87] evaluated metabolic enzyme activity using 15 drugs on the LoC system. The experiment used mathematical modeling to improve parameters, achieve accurate prediction of media sampling and evaporation in the environment with uneven media flow and concentration, and optimize the simulation process of in vitro metabolism extrapolation to in vivo metabolism. The prediction results of this in vitro-in vivo translation were closely consistent with the data observed in clinical practice, which proves that the use of the LoC platform is of great significance in quantitative metabolism studies of drug development.

### 3.2. Evaluation of Drug Efficiency and Safety

The failure of possible therapeutic candidates has been chiefly attributed to the toxicity of drugs and unidentified safety issues. Efficient identification of hepatotoxic compounds during preclinical drug development has important significance for drug development and prevention of liver injury. Prior to the invention of the LoC, primary isolated hepatocytes and immortalized cell lines were used in preclinical drug hepatotoxicity testing [84]. Common flaws in these models include rapid activity decline and decreased expression of genes and function specific to the liver [88]. Toh et al. [21] tried to combine microfluidics with hepatocytes culture in 2009. The researchers successfully cultured primary hepatocytes in multiple microfluidic channels and maintained their differentiation function to enable multiplexed drug toxicity testing in vitro. This study emphasized the important potential value of the microfluidic hepatocyte model for future drug toxicity testing, and scholars then continued their study on this basis. Bircsak et al. [20] integrated a similar 3D perfusion co-culture platform with the automatic liquid processing system to achieve high-throughput screening of drug hepatotoxicity. Researchers selected induced pluripotent stem cells derived hepatocytes (iHep) as a substitute for PHHs, and then used perfusion technology and co-culture of three kinds of liver cells to restore the liver microenvironment (Figure 3C), resulting in a long-term culture of 2–5 weeks. In each 2-lane chip, endothelial cells and differentiated THP-1 Kupffer cells were seeded into the perfusion channel, and parenchymal organ cells were seeded into the organ channel. iHeps were separated from the simulated vascular channel through a layer of endothelial cells and Kupffer cells to approach the hepatic sinuses in vascular. Albumin, urea, iHep nuclear size, and iHep viability were analyzed as indicators of hepatotoxicity. The automatic liquid processing system was used to complete automatic cell seeding, drug delivery, and culture medium collection on the platform, exhibiting stable and reliable automation performance. Selecting iHeps as a cell source was a coordination of high-throughput screening. In the future, a small number of candidate drug ingredients can be tested against a panel of models derived from dozens to hundreds of individuals.

The investigation of drug-induced liver injury (DILI) is one of the noteworthy focuses of drug development. Xiao, Rong Rong et al. [22] chose to focus on PHHs, the gold standard of drug testing, to improve and stabilize cell functions and metabolic enzyme activity by using cell-extracellular matrix (ECM) interaction. They built a collagen-based 3D PHH model on the integrated biomimetic array chip (iBAC) and compared its sensitivity of hepatotoxicity prediction with a 2D conventional culture plate. The geometry of iBAC was designed in a commercial 96-well format, making it highly compatible with commercial high-throughput devices. In comparison to 3D liver spheroids, this single-cell culture method had a more straightforward operation, lower cost, and offers excellent predictive sensitivity in drug hepatotoxicity testing. In large-scale screening tests for hepatotoxic drugs, the 3D PHH model has a high sensitivity to 122 drugs with known clinical toxicity. However, some immune-related drugs have not yet passed the test for hepatotoxicity. This aspect can be further studied through subsequent co-culture of Kupffer cells with PHHs. This model is valuable for further improvement, for example, to explore the variability of different PHH donors and become a standard evaluation tool for drug development in the future.

Another significance of LoC is to reduce the use of animal models and drug screening failures caused by physiological differences between species. Jang et al. [7] performed their studies on the human relevance of animal (rats, dogs) hepatotoxicity with 3D microfluidic liver chips. Hepatocytes and three other types of non-parenchymal cells were co-cultured in the extracellular matrix, and the culture period was sustained for 14 days. The CYP activity of the three chips was equal to that of freshly isolated primary hepatocytes or even exceeded that (when using a CYP substrate mixture), leading to effective hepatotoxic phenotype identification. This work demonstrated the value of the LoC platform in investigating the interspecific difference of hepatotoxicity, a clinical transformation of drug toxicity observed in animal studies, and its potential application in detecting the mechanism of drug-induced liver injury.

### 3.3. Multi-Organ Chips Systems

Integrating the liver with other organs of humans to form body-on-a-chip can better imitate the real physiological organ networks in vivo and incorporate the organs’ interaction factors into the research of drug ADME. Common combinations include the liver, kidney, intestine, heart, and other organs on a chip [89].

Given the significance of absorption and barrier functions of the small intestine in drug flow in vivo, Kimura et al. [90] integrated the small intestine and liver on a microfluidic perfusion chip to predict pharmacokinetics. Due to insufficient oxygen or nutrients, co-culturing cells from different organs were challenging in conventional research. Researchers made microfluidic chips with high oxygen permeability materials (polydimethylsiloxane) and embedded micropumps to simulate physiological circulation to achieve the co-culture of Caco-2 cells (small intestine), HepG2 cells (liver) and A549 cells (lung cancer) for more than three days. The model strictly reproduced physiological parameters such as organ volume ratio, blood circulation structure, and hepatic venous and arterial blood flow ratio. This work is the first trial to reproduce both blood circulation and organ volume relation in vitro.

The kidney is the main organ for eliminating drugs. The metabolism of drugs in the kidney promotes the development of drug-targeting tissue distribution systems, and drug-targeting tissue distribution can avoid some side effects of drugs. Liu et al. [23] published the first report on the pharmacokinetics of carbohydrate drugs (ginsenoside compound K, CK) using microfluidic multi-organ chips. Researchers separated the small intestine, liver, and kidney from the complex human organs to prevent the affection of complex metabolism in the human body. The several layers of chips are ordered according to drug absorption and metabolism in vivo (Figure 4A,B). HepG2 and HUVEC co-culture increased the metabolic capacity of HepG2, and dynamic perfusion culture enhanced the activity of Caco-2 cells and their tolerance to CK. Organ chips can not only simulate the physiological microenvironment but also restore the biological phenomena of organs, which is conducive to explaining the reasons behind biological phenomena. Both intestinal and vascular administration were conducted in this experiment. By observing the status of CK as the substrate for protein transporters, the differences in the uptake of CK caused by different administration methods in the intestine were explored. Moreover, the difference in uptake caused by different administration methods in the blood vessel indicates that the permeability of CK is different inside and outside the blood vessel. The results of absorption, metabolism, and toxicity tests of this system were consistent with those of existing traditional platforms, demonstrating its potential application in carbohydrate drug research.

Another crucial topic that needs to be investigated in drug metabolism studies is drug–drug interactions (DDIs), which occur when the ADME properties of one pharmacological are affected by the concurrent administration of another drug component. In clinical practice, tumor patients are prone to DDI due to their frequent medication use and narrow treatment index of anticancer drugs. Therefore, Lohasz et al. [92] selected human liver and tumor microtissues as the culture in a multiple-organ system for simultaneous drug detection. Researchers constructed a gravity-driven microfluidic platform to maintain or even increase the activity of CYP subtypes. The hepatocytes used were derived from 10 different donors to form human liver microstructures to limit variability between donors. Two anticancer prodrugs with similar structures, cyclophosphamide (CP) and ifosfamide (IFF), were selected for the experiment and used in combination with ritonavir (RTV), which has an inhibitory effect on CYP subtypes. The metabolites in the model indicated that co-administration of IFF and RTV may result in the conversion of a significant portion of IFF into neurotoxic/nephrotoxic metabolites. The combination of IFF and CP has the probability of causing severe side effects in cancer patients. This DDI detection results in the chip being consistent with known phenomena. This excellent capture and quantification ability for the DDI phenomenon is believed to significantly improve the early drug testing process in the preclinical stage.

In addition to detecting the DDI phenomenon directly, we also seek to comprehend the pharmacokinetics of DDI in detail, developing more practical methods for DDI prediction. Kenta et al. [91] combined the PK/PD mathematical model with a multiple-organ-on-chip system (MOoC) to evaluate the value of organ chips in the risk assessment of concomitant drugs (Figure 4C,D). Researchers used the result parameters of liver metabolism of a single drug in MOoC to construct PK/PD model and then carried out a concomitant administration experiment in MOoC. The experimental results were highly consistent with the simulation results of the PK/PD model, which proved the effectiveness of this combination in predicting DDI. Table 1 summarizes some applications of LoC in drug metabolism, drug toxicity test, drug interaction, and other studies.

## 4. Conclusions and Future Perspective

The advances described above demonstrate that drug discovery using microfluidic platforms can be more productive than traditional methods. Both single liver-on-a-chip and multi-organs-on-chip models have numerous advantages for the drug metabolism simulation and toxicity test stage of preclinical studies. Compared with 2D culture and static culture, a dynamic 3D culture platform with microfluidic significantly extends the duration of cellular activity in vitro, providing a solid foundation for drug testing. Adding related chemicals increases the activities and drug sensitivity of cells or tissues, sometimes even outperforming newly obtained hepatocytes. The LoC platform is more effective than most in vitro physiological systems at simulating the natural liver environment, leading to an accurate recreation of the results of many previous drug tests [93,94,95].

In contrast to organoids, microfluidic systems will not change shape over time and have strong repeatability and controllability, making them ideal for large-scale preclinical drug screening and testing [96]. In addition to independent research, liver chips provide more precise physiological parameters for existing mathematical models. They can also be integrated with laboratory automation equipment, making continuous progress toward high throughput screening.

At present, however, liver chips cannot replace animal models completely. Researchers often use organ chips to reproduce key drug-related events when they observe them in animals. The controllability and reproducibility of liver chips are very useful for studying the details of candidate drug compounds and long-term repeated treatment issues. Although animal models must deal with the cost problem, the complex components and manufacturing technology required by liver chips cannot avoid the cost issue. Complexity limits the large-scale manufacturing of liver chips, increases the cost of operation training, and limits the development of high-throughput automation. Meanwhile, there are also crucial problems in the physiological simulation of liver microarray, such as the cell source being closely related to the physiological relevance of the physiological model. PHHs are still the best choice at present. The microfluidic technology can prolong their active time in vitro, but the donor supplies are limited. The current trend is to replace PHHs with inexpensive, highly differentiating iPSCs [97,98,99,100,101]. The differentiation characteristics of iPSCs can also provide convenience in personalized drug research. However, using iPSCs also poses challenges. The process of cell differentiation is complex, not only with many influencing factors but also taking a long time, and the complexity of the operation needs to be reduced. In addition, cells derived from iPSCs are not entirely consistent with native liver cells, and their structure and function are immature [102]. Although it is a good idea to use relatively simple models to study complex phenomena and problems, there is still much room for improving the physiological relevance of models [103].

In addition, this new technology with complex functions is difficult to match the commonly used microscope imaging methods. Manual positioning and manipulation of organ chips for image acquisition are variable and time-consuming, and the amount of data is limited [104]. Nevertheless, this method is often used for expensive and professional equipment, such as two-photon microscopy [105]. Semi-automatic and automatic imaging and quantification software can improve the quantity and quality of data. However, it is also necessary to overcome the problem that the current automatic imaging technology does not match the uniqueness of the liver chip platform. In 2019, a study described the automatic imaging program of the liver-on-a-chip [106], and some other scholars tried to use 3D printing technology to design and create a chip stent for this organ chip later [107]. However, many current imaging methods are still incompatible with organ chips, such as holographic optical fluid microscopy, which provides low resolution and does not meet the micro resolution required by organ chip platforms [108].

In the long run, compatibility with laboratories and automatic high-throughput screening are the constant goals of LoC technology in drug development. Currently, the maximum number of organ chips in an MPS device is only 96. The throughput of this scale will inevitably pose obstacles to high-throughput screening when subsequent research is accelerated [109]. Miniaturization and automation of organ chips are ongoing goals. In the future, 96-, 384-well, or more porous microtiter plates should be further developed to be compatible with existing pharmaceutical laboratory automation equipment. At the same time, we can further explore the manufacturing process, cell source, culture conditions, and other factors to reduce the production cost and operational complexity. For example, PDMS, widely used in manufacturing microfluidic chips, has problems to solve. The high hydrophobicity of PDMS makes it difficult for cells to adsorb without modification, and it also allows drug molecules to gather and be absorbed in unpredictable ways. Deguchi, Sayaka et al. [110] evaluated the absorption of 12 hydrophobic small molecule compounds on PDMS plates. The continued search for alternative materials with oxygen permeability, optical transparency, and non-absorbability is also one of the future research directions in this field. We should also actively integrate microfluidic technology with research methods in other fields to broaden its scope of application, such as mathematical model analysis. The imaging technology of LoC is also worthy of continuous attention. Traditional fluorescence imaging methods are mainly used to confirm the cell activity in the chips. Imaging methods can be mended by improving the resolution, reducing the cost, or facilitating transportation. It should also be noted that liver chips may show significant differences and inconsistencies between manufacturing batches, laboratories, or even manufacturers in the same group. Especially for multi-organ chips, different organ models in vitro have different cultivation and manufacturing technologies. A standardized system must combine them well to reduce adverse factors in preclinical drug testing. Therefore, the commercialization of chips also deserves attention. As a result, companies produce more reliable and consistent equipment, which will significantly increase the availability of laboratories and industries worldwide.

## Figures and Tables

**Figure 1 pharmaceutics-15-01300-f001:**
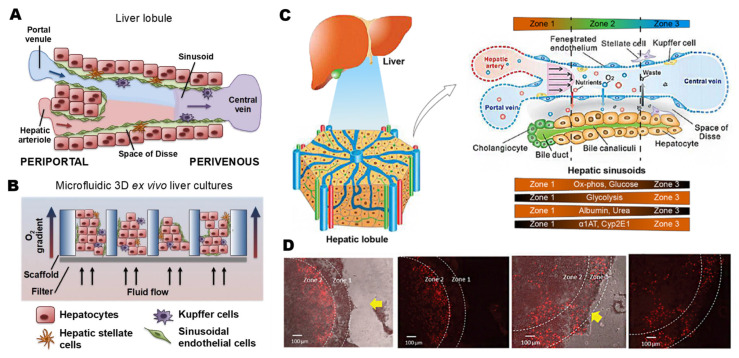
(**A**) Cell distribution structure in a liver lobule. Double blood supply of hepatic arteriole and portal vein enters the liver’s portal triad region and the central vein. (**B**) Three-dimensional liver tissue diagram in LoC. The fluid flows through the pores on the scaffold, providing the biochemical environment and mechanical stimulation. (**C**) Microstructure of liver. The liver lobule is hexagonal, with a 1 mm diameter and a 2 mm thickness, functionally divided into three regions. (**D**) Example of zonal heterogeneity of hepatotoxicity. The yellow arrow shows the flow direction. (**A**,**B**) reproduced with permission of [34], Copyright © 2017. Published by Elsevier. (**C**,**D**) reproduced with permission of [29], Copyright © 2019. Published by MDPI.

**Figure 2 pharmaceutics-15-01300-f002:**
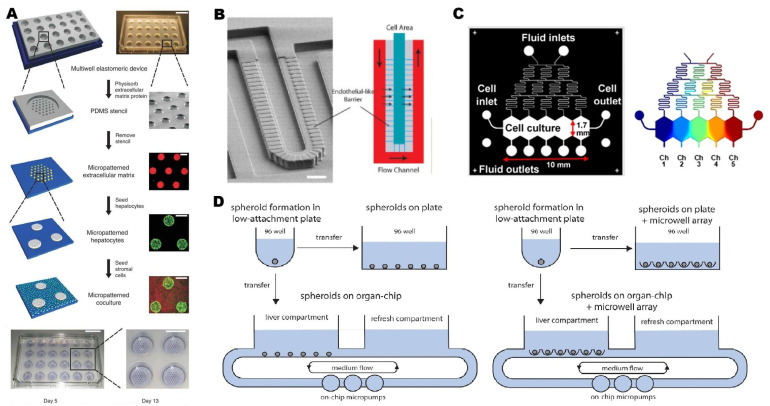
Different microfluidic models are used to construct liver-on-a-chip. (**A**) A 24-well PDMS mold with through holes at the bottom. Hepatocytes (labeled green) selectively adhered to the collagen structure, clustered together, and then surrounded by mouse 3T3-J2 fibroblasts (labeled orange). (**B**) Microfluidic endothelial-like barrier. The barrier consists of parallel channels connecting the cell culture zone with the external flow channel. (**C**) Simulation of liver lobules in Christmas tree structure, with gradient generator. (**D**) Spheroid culture. Cultured spheroids in an ultra-low-attachment plate, transferred them to a 96-well plate or liver chip and compared their performance. (**A**) reproduced with permission of [11], Copyright © 2008. Published by Nature Publishing Group. (**B**) reproduced with permission of [78], Copyright © 2007. Published by Wiley Periodicals, Inc. (**C**) reproduced with permission of [79], Copyright © 2018. Published by Nature Publishing Group. (**D**) reproduced with permission of [80], Copyright © 2022 Published by Elsevier.

**Figure 3 pharmaceutics-15-01300-f003:**
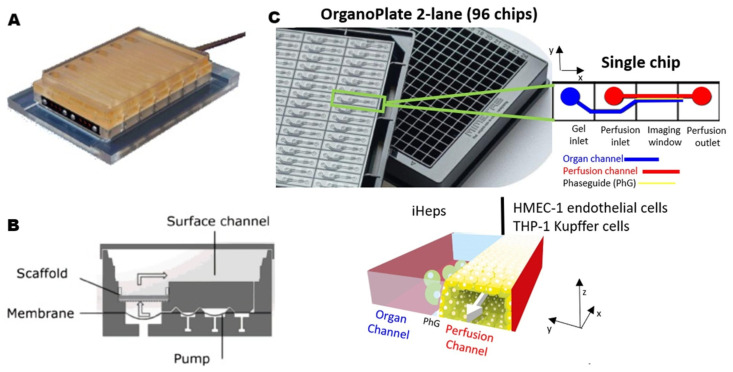
Example of 3D perfusion platforms. (**A**) Twelve fluid-isolated bioreactors with connected cell culture plate (yellow) and pneumatic plate (gray). (**B**) Cross section diagram of the bioreactor, the membrane separates the upper cell culture plate from the lower pneumatic plate, and the micropump provides power for medium circulation in the channel. (**C**) OrganoPlate LiverTox model. The iHeps differentiated in 2D culture are harvested and seeded in OrganoPlate 2-lane through a liquid processor. Perfusion and organ channels were separated by phase guidance (PhG). (**A**,**B**) reproduced with permission of [16], Copyright © 2016. Published by The American Society for Pharmacology and Experimental Therapeutics. (**C**) reproduced with permission of [20], Copyright © 2021. Published by Elsevier.

**Figure 4 pharmaceutics-15-01300-f004:**
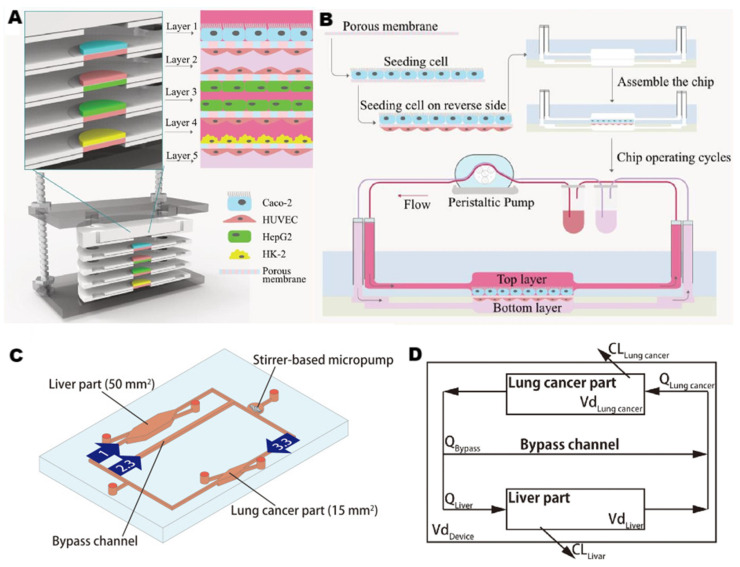
Example of multi-organ chips platforms. (**A**) Multi-organ chips based on liver, intestinal, vascular, and kidney chips. Caco-2, HUVEC, HepG2, and HK-2 occupy one floor from top to bottom (**B**) Taking a two-layer chip assembly as an example, the system was built in the order of inoculating cells to a porous membrane, providing a cell culture medium and assembling chips. The peristaltic pump provides power for fluid. (**C**) The liver and lung cancer parts on chips are connected through a microchannel. Channel is also equipped with a stirrer-based micropump to provide power. The value in the arrow is the flow ratio. (**D**) Concept diagram of PK model based on the multi-organ chip. (**A**,**B**), reproduced with permission of [23], Copyright © 2020. Published by Elsevier. (**C**,**D**) reproduced with permission of [91], Copyright © 2020. Published by AIP Publishing.

**Table 1 pharmaceutics-15-01300-t001:** Summary of the application of LoC in drug development.

Application	Characteristic	Cell Types	Drugs Involved	Analysis	Ref.
Drug metabolism	3D perfusion, co-culture	hepatocytes, Kupffer cells	hydrocortisone	the anti-inflammatory effect of glucocorticoids on liver cultures	[18]
Drug metabolism	3D perfusion, co-culture	PHHs, Kupffer cells	tocilizumab	dual regulation of inflammatory factors and therapeutic antibodies	[16]
Drug metabolism	3D perfusion, co-culture, liver–small intestine	Caco-2, HepG2, A549	epirubicin (EPI), irinotecan (CPT-11), and cyclophosphamide (CPA)	reproduce both blood circulation and organs volume relation in vitro	[90]
Drug metabolism	3D perfusion, co-culture, liver–small intestine–kidney	Caco-2, HepG2, HUVEC, HK-2	ginsenosides compound K (CK)	the pharmacological investigation of carbohydrate drugs	[23]
Population variability of liver drug metabolism	3D perfusion	PHHs from five different donors	phenacetin, diclofenac, lidocaine, ibuprofen, propranolol, and prednisolone	in vitro assessment of population variability in drug metabolism	[19]
Drug toxicity	3D microfluidics	hepatocytes (rats)	acetaminophen, diclofenac, quinidine, rifampin and ketoconazole	multiplexed testing	[21]
Drug toxicity	3D perfusion, co-culture	iPSC-derived hepatocytes, endothelial cells, Kupffer-like immune cells	troglitazone, a library of 159 compounds	high throughput screening	[20]
Drug toxicity	collagen-based 3D model, integrated biomimetic array chip, cell–extracellular matrix interaction	PHHs	122 clinical drugs evaluated for liver toxicity	large-scale hepatotoxicity screening	[22]
Drug toxicity	3D microfluidics, co-culture, cell–extracellular matrix interaction	hepatocytes, sinusoidal endothelial cells, Kupffer cells, hepatic stellate cells (humans, rats, and dogs)	bosentan, analgesic acetaminophen (APAP), methotrexate (MTX), fialuridine (FIAU), a Janssen proprietary compound (JNJ-2)	human and cross-species drug toxicities	[7]
Drug-drug interactions	gravity-driven microfluidic, liver–intestines	PHHs, HCT116	cyclophosphamide, ifosfamide, ritonavir	quantization of the impact of other drugs on the efficacy of anticancer drugs	[92]
Drug-drug interactions	3D microfluidics, liver-lung	HepG2, A549	simvastatin, ritonavir, CPT-11	quantization of the impact of other drugs on the metabolism of anticancer drugs	[91]

## Data Availability

Not applicable.

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
