# Peer review of "Microfluidic Liver-on-a-Chip for Preclinical Drug Discovery"

_pharmaceutics, 2023, doi:10.3390/pharmaceutics15041300_

Round 1

Reviewer 1 Report

This review focused on the microfluidic liver-on-a-chip for drug discovery, which summarized the current methods of constructing LOC and the pharmacological and toxicological application. This work contributes to the study of LOC and can help researchers to quickly understand the current status. The following are some suggestions:

1. The language of the manuscript needs further improvement.

2. Microfluidic technology is the most important to construct LOC, however, the authors are weak on the elaboration of microfluidic techniques (2.2.2). A more detailed review of current research on constructing LOC based on microfluidics is suggested to classify, summarize and discuss.

3. In the Future Perspective section, it is suggested that the authors add the outlook on LOC construction and application technology. The current discussion is rather general.

Author Response

As attached

Reviewer 2 Report

This article sorts out the related research on Liver-on-a-Chip, and the structure of this manuscript is well-organized and easy to follow. For the content of this manuscript, the number of integrated figures and the summary table seems insufficient for a review article. Besides, there is not much discussion on technical and performance comparisons. The most important point is that there have been many review articles on this topic in recent years [1-5]. Although this article cited part of above-mentioned articles, the sentences referring to these articles seem to be less relevant to the articles. The feature of this review article (perhaps a discrepancy in drug testing?) could be more clearly defined in the revised version.

[1] Ehrlich, A., Duche, D., Ouedraogo, G., & Nahmias, Y. (2019). Challenges and opportunities in the design of liver-on-chip microdevices. Annual review of biomedical engineering, 21, 219-239.

[2] Özkan, A., Stolley, D., Cressman, E. N., McMillin, M., DeMorrow, S., Yankeelov, T. E., & Rylander, M. N. (2020). The influence of chronic liver diseases on hepatic vasculature: a liver-on-a-chip review. Micromachines, 11(5), 487.

[3] Deguchi, S., & Takayama, K. (2022). State-of-the-art liver disease research using liver-on-a-chip. Inflammation and Regeneration, 42(1), 1-10.

[4] Dalsbecker, P., Beck Adiels, C., & Goksör, M. (2022). Liver-on-a-chip devices: the pros and cons of complexity. American Journal of Physiology-Gastrointestinal and Liver Physiology, 323(3), G188-G204.

[5] Yang, Z., Liu, X., Cribbin, E. M., Kim, A. M., Li, J. J., & Yong, K. T. (2022). Liver-on-a-chip: Considerations, advances, and beyond. Biomicrofluidics, 16(6), 061502.

Author Response

As attached

Reviewer 3 Report

General comments

This manuscript reviews, in a non-critical manner, a wealth of publications in the microfluidic liver on a chip area. It is nicely illustrated and presented in a logical manner and easy to read. The application focus for drug discovery is quite weak, considering that the title includes “…for preclinical drug discovery”. There does not seem to be much in the way of differentiation points between this review and the many other MPS reviews published in the last couple of years. Suggestions are made in the comments for ways to strengthen these elements.

Major comments

(1) The focus on preclinical drug discovery does not seem to be very strong. Suggest to the authors to consider the applications of liver-on-a-chip (and other systems incorporating this such as gut-liver-on-a-chip) to preclinical drug discovery. What assays would be run using a LoC system? How would these improve drug candidate selection? What validation is needed in order for the system to be adopted? Clues for this can be gained from the adoption of 2D long-term hepatocyte co-culture systems such as hepatopac as well as recent publications where hepatocyte spheroids have been applied to ADME questions.

Ideas for resources:

a) IQ MPS working group position papers and discussion reports (e.g. DOI: 10.1039/c9lc01168d; DOI 10.1039/c9lc00857h; doi:10.14573/altex.2112203)

b) Pharmaceutical industry publications on long-term hepatocyte applications (https://doi.org/10.1124/dmd.120.090951; https://doi.org/10.1124/jpet.117.245712; DOI: 10.1208/s12248-016-0019-7; https://doi.org/10.1016/j.xphs.2018.03.001; https://doi.org/10.1124/dmd.120.000133; http://dx.doi.org/10.1124/dmd.115.067173; http://dx.doi.org/10.1124/dmd.113.055897; DOI: 10.1208/s12248-020-00482-9)

c) Liverchip and gut-liver chip for ADME explorations published in 2022: (DOI: 10.1039/d1lc01161h; DOI: 10.1039/d2lc00276k)

(2) How is this manuscript differentiated from the many other MPS technologies reviews? The literature is quite saturated with MPS and organ-on-a-chip review articles. The current article feels like a useful source of links to important materials but I don’t feel that those working in the field will currently learn something new or see things from a different perspective as a result of reading this article. Suggest to the authors that some more specific focus / more clear messaging is needed in order to make their article memorable.

Minor Comments

(1) I consider the statement in Lines 118-119: “ Currently, the widely used in vitro liver model to analyse drug metabolism is the traditional 2D culture of hepatocytes.” to be outdated and misleading. Within the industry people either make use of suspension cultures of cryopreserved hepatocytes or of more sophisticated long-term hepatocyte co-culture systems. Traditional 2D cultures are probably only really used for induction testing in the ADME sphere nowadays. Request to update this statement or provide up-to-date references to substantiate it.

(2) Lines 198-201 “The initial static culture platforms inspired perfused devices, although they could not meet the requirements of shear stress. For example, Khetani et al.[76] designed a platform which showed the importance of co-culture and spatial structuring. However, these plat-forms are not versatile enough, suitable for research with a given purpose.

This gives the impression that the micropatterned co-culture system from the Khetani reference is unsuitable for research purposes whereas this is a widely adopted platform that is used much more in the drug metabolism area of drug discovery than any of the microfluidic liverchip systems. Request to re-word to remove this potential for misunderstanding.

Author Response

As attached.

Round 2

Reviewer 2 Report

The revised manuscript has addressed all of the concerns we raised. The manuscript is now suitable for publication.